# Alginate-Based Hydrogel as Delivery System for Therapeutic Bacterial RNase

**DOI:** 10.3390/polym14122461

**Published:** 2022-06-16

**Authors:** Liliya R. Bogdanova, Pavel V. Zelenikhin, Anastasiya O. Makarova, Olga S. Zueva, Vadim V. Salnikov, Yuriy F. Zuev, Olga N. Ilinskaya

**Affiliations:** 1Kazan Institute of Biochemistry and Biophysics, FRC Kazan Scientific Center of RAS, Kazan 420111, Russia; chemli@mail.ru (L.R.B.); tat355@mail.ru (A.O.M.); vadim.salnikov.56@mail.ru (V.V.S.); 2Institute of Fundamental Medicine and Biology, Kazan Federal University, Kazan 420008, Russia; pasha_mic@mail.ru; 3Department of Physics, Kazan State Power Engineering University, Kazan 420066, Russia; ostefzueva@mail.ru

**Keywords:** drug delivery, alginate microspheres, divalent cations, toxicity, enzymatic therapy, ribonuclease, antitumor properties, antiviral activity

## Abstract

To deliver therapeutic proteins into a living body, it is important to maintain their target activity in the gastrointestinal tract after oral administration. Secreted ribonuclease from *Bacillus pumilus* (binase) has antitumor and antiviral activity, which makes it a promising therapeutic agent. This globular protein of small molecular weight (12.2 kDa) is considered as a potential agent that induces apoptosis of tumor cells expressing certain oncogenes, including colorectal and duodenum cancer. The most important problem of its usage is the preservation of its structure and target activity, which could be lost during oral administration. Here, we developed alginate microspheres reinforced with divalent cations and analyzed the enzyme release from them. Using methods of scanning electron microscopy, measurements of fluorescence, enzyme catalytic activity, and determination of viability of the duodenum adenocarcinoma tumor cell line, we characterized obtained microspheres and chose calcium as a biogenic ion-strengthening microsphere structure. Among such modified additivities as beta-casein, gelatin, and carbon nanotubes introduced into microspheres, only gelatin showed a pronounced increase in their stability and provided data on the prolonged action of enzyme release from microspheres into tumor cell culture medium during 48 h in an amount of about 70% of the loaded quantity.

## 1. Introduction

The hydrogels based on natural biopolymers of polysaccharide origin are promising innovative biomaterials for drug encapsulation and delivery [1,2,3]. Biocompatibility of hydrogels with human intercellular matrix due to the similarity of their structure and physicochemical properties determine the prospects for their use in medicine [4,5]. The most technologically advanced hydrogels are biopolymers organized into a three-dimensional physical network through the mechanical interweaving of polymer molecules and intermolecular interactions, including ionic bridges, hydrogen bonds, and hydrophobic forces [6,7,8]. The polysaccharide-based materials like mesoporous hydrogels display original physicochemical and biological properties [9,10], providing a method for novel biomedical applications, such as drug delivery [11,12,13,14], tissue engineering, and regenerative medicine [15,16], biosensing, imaging, and molecular diagnostics [17,18,19].

Natural polysaccharides demonstrate low toxicity, improved biocompatibility, and stability of engineered complexes under physiological conditions. However, some of the designed materials are often of complicated composition, thus acquiring toxic properties and losing specific functions. The aim of our work was to design microspheres based on the ion-reinforced sodium alginate, to analyze their physicochemical characteristics and toxic properties, to load microspheres with RNase, which has antitumor and antiviral activities [20,21,22,23,24], and to record the dynamics of enzyme release.

Over the past decades, researchers have focused on some cytotoxic RNases as potential therapeutic agents. Many of RNases attack selectively malignant cells, triggering apoptotic response, and therefore they are considered as alternative chemotherapeutic drugs. Unlike mammalian RNases, being potently inhibited by ubiquitous cytoplasmic eukaryotic RNase inhibitor, the guanyl-preferring secreted RNase from *Bacillus pumilus* (binase) manifests its catalytic activity towards available RNA molecules inside cells as a result of their high stability and lack of susceptibility to RNase inhibitors. For loading into microspheres, we chose binase possessing antiviral [25,26,27] and antitumor [28,29,30] properties. The killer strategy of cytotoxic RNases includes the hitting of its main target, RNA, but does not exclude additional ravages leading to the cell death [28]. It was shown that binase is able to inhibit the replication of MERS-CoV and of the low-pathogenic human coronavirus 229E in cell culture, decreasing titers of both viruses at non-toxic concentrations, reducing accumulation of viral subgenomic RNAs, viral nucleocapsid protein, and non-structural protein 13 [25]. Moreover, binase acts as an antiviral agent at the level of whole animal organism infected by mammalian orthoreovirus 1 strain Lang [27]. The selective tumor cytotoxicity of binase is determined not only by its catalytic activity and the ability to avoid inhibition by mammalian RNase inhibitor, but also by its ability to interact with certain cellular components and oncogenic proteins [29]. Considering our obtained results, it is necessary to add a change in the profile of cellular microRNAs. The analysis of binase-susceptible miRNAs and their regulatory networks showed that the main modulated events were the transcription and translation control, the cell cycle, cell proliferation, adhesion and invasion, apoptosis, and autophagy, as well as some other tumor-related cascades, with an impact on the observed anti-tumor effects [30]. These data provide a reasonable idea of the use of binase as an antitumor agent. Despite the enzyme resistance to proteolysis, its oral administration in encapsulated form will create favorable conditions for gradual release in the gastrointestinal tract. Alginate is known as biocompatible harmless substance [8,13] and is used in medicine to bind and remove from the body the metabolic by-products, salts of heavy metals, and radionuclides as well as to reduce stomach acidity (for example, Antacid from USA manufacturers, NutraScience Labs, Pharbest Pharmaceuticals, NY). Since alginate neutralizes the acidic environment, it will contribute to a better manifestation of binase catalytic activity, whose pH-optimum is 8.5. That is why we set as the goal of this work to include binase in the alginate-based hydrogel.

The employment of polysaccharides in real biomedical practice often requires additional modifications of engineered systems. One of the accepted ways to improve functional properties of desired systems is the use of such nanomaterials as carbon nanotubes (CNTs) [31].

## 2. Materials and Methods

### 2.1. Chemicals

Sodium alginate (A2033) and κ-carrageenan Type I purchased from Sigma-Aldrich, USA, were used to prepare hydrogels. Inorganic salts: calcium chloride (CaCl_2_), zinc sulfate heptahydrate (ZnSO_4_·7H_2_O), barium chloride dihydrate (BaCl_2_·2H_2_O), manganese chloride tetrahydrate (MnCl_2_·4H_2_O), copper sulfate pentahydrate (CuSO_4_·5H_2_O) from Tatchemproduct, Russia, and nickel sulfate hexahydrate (NiSO_4_·6H_2_O) from Sigma were used to prepare microspheres. Multiwalled Taunit carbon nanotubes (CNTs) with an average outer diameter 20–50 nm and the length not exceeding 1000 nm were obtained from NanoTekhCentr, Tambov, Russia. For dispersing CNT in water, the gelatin from porcine skin type A (Sigma, Missouri, USA, G2500) was used. In some experiments, beta-casein from Lactalis (Lactalis, Laval, France) was used. Water purified with the “Arium mini” ultrapure water system (Sartorius, Goettingen, Germany) was used to prepare all solutions. For Tris-HCl buffer tris(hydroxymethyl)aminomethane from Helicon and hydrochloric acid from Tatchemproduct, Russia, was implicated.

### 2.2. Ribonuclease

The guanyl-specific RNase from *B. pumilus 7P*, hereinafter, binase (12.2 kDa, 109 amino acid residues, pI 9.5), was isolated from the culture fluid of Escherichia coli BL21 carrying the pGEMGX1/ent/Bi plasmid, according to Dudkina et al. [32]. The catalytic activity against synthetic substrates [33] and high-polymeric yeast RNA was already known [34]. The maximum activity determined according to Kolpakov and Ilinskaya [35] was 14,000,000 U/mg at pH 8.5. An activity unit is the amount of enzyme capable to increase the extinction at 260 nm of acid-soluble products of RNA hydrolysis to 1 unit per min at 37 °C and pH 8.5.

### 2.3. Preparation of Initial Solutions

The concentrated solutions of sodium alginate (4 wt.%) were prepared by dissolving polysaccharide samples in 50 mM Tris-HCl buffer pH 7.4, preliminary swelling at room temperature, and subsequent heating to 70 °C; the solution of gelatin (1 wt.%) was prepared similarly by heating at 45 °C. To increase the retention of enzyme in microspheres, casein was used at final concentrations 600–800 µg/mL.

Initially, the enzyme-carrying system (ECS) consisted of 0.5 mL of sodium alginate and 0.5 mL of enzyme. The modified ECS contains 0.5 mL of sodium alginate, 0.45 mL of enzyme solution (11 mg/mL), and 0.05 mL of a gelatin solution. The resulting solutions were sonicated in Bandelin SONOREX TK52 ultrasonic bath (Germany) for 30 min at 35 kHz, 100 W, 35 °C. The polysaccharide concentration in ECS was 2 wt.% for sodium alginate, and the concentration of enzyme was 5 mg/mL.

Three mg of CNTs was dispersed in 1 mL of gelatin solution and sonicated in Bandelin SONOREX TK52 ultrasonic bath for 60 min at 35 kHz, 100 W, and 45 °C. To separate undispersed CNTs the samples were centrifuged (ELMI CM-50M Fugamix™ microcentrifuge) for 10 min at 10,000× *g*.

### 2.4. Preparation of Polysaccharide Microspheres

To prepare microspheres from alginate hydrogel, 0.5 mL of 2 wt.% alginate solutions were added dropwise under constant stirring (500 rpm) to 1.5 mL of 1M solutions of Ca, Mn, Ni, Cu, Zn, and Ba salts with a medical syringe (needle diameter of 0.63 mm). The prepared microspheres were left in a salt solution for 10 min and then washed three times with 50 mM Tris-HCl buffer pH 7.4. Each binase-loaded microsphere had a volume of 4–4.2 µL and included about 20 µg of the enzyme.

### 2.5. Enzyme Release from Hydrogels and RNase Content in Microspheres

To assess the holding capacity of hydrogel, to the adjusted volume of freshly prepared ECS, 4 mL of 50 mM Tris-HCl buffer (pH 7.4) or PBS (pH 7.2) was added. Enzyme (binase) release was determined spectrophotometrically after 4 h of incubation at room temperature. The extinction coefficients of enzyme were determined in an independent experiment as 27,000 M^−1^cm^−1^ (280 nm). Enzyme release from beta-casein- or gelatin-containing alginate microspheres was measured by the RNase catalytic activity.

To determine activity of RNase, loaded into microspheres without gelatin, the protein concentration was measured spectrophotometrically at 280 nm after complete lysis of 10 capsules in 10 mL PBS. Protein concentration in 1 mL solution corresponded to content of RNase in one microsphere. For gelatin-containing microspheres, catalytic activity of RNase was determined in this solution, and the activity units were recalculated for content of protein in one microsphere.

### 2.6. Cytotoxicity Assay

Cytotoxicity of alginate microspheres fortified with different cations was analyzed using the MTT-test with 3-(4,5-dimethylthiazol-2-yl)-2,5-diphenyltetrazolium bromide (Sigma-Aldrich, Missouri, USA) following the manufacturer’s instructions. Duodenum adenocarcinoma HuTu-80 cells were obtained from the Russian cell culture collection of vertebrates (Saint Petersburg, Russia). Cells were cultured at 37 °C in a humidified atmosphere with 5% CO_2_ in DMEM (Gibco) supplemented by 10% fetal calf serum (HyClone), 100 U/mL penicillin, and 100 U/mL streptomycin. The experiment was carried out in 24-well plates, in which 1.5 × 10^5^ cells per well containing 500 μL medium were seeded 24 h before the experiment. After confluence, medium was replaced with the fresh one, and alginate microspheres were added to wells in cultural inserts with 0.4 µm pores (SPL, Pocheon-si, South Korea). The amount of protein in 5 microspheres corresponded to positive control concentration (100 μg of enzyme in 1 mL medium). After 48 h, the cell viability was measured. As negative control blank ECS without enzymes was used.

### 2.7. Scanning Electron Microscopy

The scanning electron microscopy (SEM) was performed to control the structure of freeze-dried (Martin Christ) samples of hydrogel microspheres (Merlin autoemission scanning electron microscope, Carl Zeiss, Oberkochen, Germany).

### 2.8. Fluorescence Measurements

Fluorescence spectroscopy of tryptophan residues [36] was used to study possible structural alterations of protein (binase) encapsulated in hydrogel microspheres. Fluorescence spectra were obtained with the Fluorate-02-Panorama spectrofluorimeter (Lumex, Saint-Petersburg, Russia). Emission spectra were recorded between 310 and 400 nm, every 1 nm, with excitation wavelength set at 295 nm. All measurements were performed in quartz cells with the optical pathway 1 cm.

### 2.9. Water Swelling Capacity of Microspheres

The swelling rate was determined by a known method [37,38,39]. A weighted quantity of lyophilized microspheres was submerged into 50 mM Tris-HCl buffer (pH 7.4) at a fixed temperature (25 °C) in a shaker (Mikrowstrzasarka ML-1) at shaking speed 100 rpm. Microspheres were removed from solution after 24 h; excess water was wiped off and the weight of swollen microspheres was determined. The swelling rate was determined using the following expression:(1)$$Swelling\;\left(\%\right)=\frac{m_{s}-m_{d}}{m_{d}}\times 100,$$
where *m_s_* and *m_d_* are the masses of water-swollen and dry microspheres, respectively.

## 3. Results

### 3.1. Structural Features of Alginate Microspheres and Water Swelling

We created microspheres based only on alginate, as well as those with various additives—casein, gelatin, and carbon nanotubes—to enhance enzyme retention and improve microsphere stability. The scheme of alginate jelling and example of obtained alginate microspheres are shown in Figure 1. It is known that alginate has an extended conformation of linear polymeric chain, composed of (1→4)-linked α-L-guluronic and β-D-mannuronic residues of varying sequence depending on the natural source of algae origin [40]. In general, alginates have stiff molecules due to the rigid six-membered sugar rings and restricted rotation around the glycosidic linkage. In addition, the electrostatic repulsions between charged groups in alginate chain contribute to the rigidity of their chains. The extended rigid worm-like molecular chains of alginate could not form the effective chain entanglements in aqueous solution. Gelation of alginate takes place under the mild conditions with the temperature independent sol-gel transition. To strengthen the alginate hydrogel, we used several metal cations. Divalent cations replace univalent Na^+^ and induce the chain–chain association by means of physical cross-linkage of guluronic acid blocks of neighboring alginate molecules (Figure 1), known as the “egg-box model” [41].

The variation of divalent cations added to sodium alginate results in the different inner structure of hydrogels. In the present study the salts of Ba, Ca, Zn, Cu, Ni, and Mn, with decreasing ion radii [42]: 0.135 nm (Ba), 0.099 nm (Ca), 0.074 nm (Zn), 0.073 nm (Cu), 0.070 nm (Ni), and 0.067 nm (Mn) were used for polysaccharide jelling. It was shown that the type of divalent cation affected the arrangement of polysaccharide chains and the size of pores in hydrogel (Figure 2). To characterize the internal structure of cation–alginate hydrogels the prepared microspheres were freeze-dried, and intrinsic porous structure was examined using scanning electron microscopy. In gels with alkaline earth cations Ba^2+^ and Ca^2+^ (Figure 2A,C) the size of the rhombic cells was practically independent of ionic radius within the interval 50 ÷ 70 × 25 ÷ 40 µm. For gels based on transition metals ions Mn^2+^, Zn^2+^, Cu^2+^, and Ni^2+^ (Figure 2B,D–F) the cell size was much smaller and slightly decreased with ionic radius decreasing within the range 20 ÷ 30 × 15 ÷ 25 µm. The shape of cells was rather cubic than rhombic. In further studies, we settled on the biogenic stabilizing ions, the most widely represented in living organisms, namely Ca^2+^ and Mn^2+^.

The water swelling of the microspheres is a result of solvent absorption by the hydrogel structure. This parameter correlates with stability of hydrogel. Table 1 shows the value of water swelling of microspheres stabilized with various cations. According to our data, the lowest water swelling was observed for microspheres stabilized with Ca^2+^ and Zn^2+^.

### 3.2. Influence of Alginate on Enzyme Structure

To confirm the preservation of enzyme in microspheres, we controlled the influence of alginate environment on the binase structural stability. The loading of binase in alginate hydrogel can influence enzyme structure and, thus, its catalytic and pharmacological properties. To evaluate possible alterations in binase structure, the tryptophan fluorescence was used, which is rather sensitive to the surrounding of this residue. The binase molecule consists of 109 amino acids including three tryptophans (Figure 3C), with their fluorescence being a template of changes in protein structure under the influence of the environment. For analysis of obtained fluorescence spectra, we used a model of discrete states, according to which all tryptophan residues in proteins are divided in five classes with strictly defined fluorescence wavelength in dependence on local microenvironment of tryptophan in protein [43,44]. Using a well-known algorithm for decomposition of tryptophan fluorescence spectra of proteins [45], we determined that binase spectrum consists of two components. An example of decomposition of total fluorescence spectrum into individual components is shown in Figure 3A. A weak progressive shift of the I class tryptophan maximum was determined with increase in alginate concentration, which showed slight compaction of protein tertiary structure under the contacts with the alginate matrix. The absence of evident changes in position of the III class tryptophan maximum (Figure 3B) signifies the absence of protein denaturation in polysaccharide environment. These results show the lack of considerable changes in protein structure under the influence of the studied hydrogel environment, which one has to keep in mind when analyzing protein functional activity in the encapsulated state.

### 3.3. Alginate Microsphere Stability and Enzyme Release

We found that all used divalent metal cations could induce the gelation of alginate solution and stabilize the formed structures. In the present study, the microspheres about 2 mm in diameter were obtained. Because the high concentrations of ions and their considerable release from microcapsules can have a negative effect on the cells and the body, we tried to reduce the concentration of stabilizing ions. However, it turned out that the concentration of divalent cations is critical for stability of microcapsules. Unfortunately, the halving of their concentration led to fast destruction of microcapsules in 50 min in Tris-HCl buffer, and in 25 min in the PBS one (Figure 4), which does not meet objectives for providing of the RNase prolonged action. These data indicate the need to use 1M solutions of divalent metal salts and the presence of destabilizing effect of phosphate ions on microcapsules. We found that half of microspheres created from sodium alginate without additives were destroyed in PBS after 4 h with complete destruction in 24 h (Figure 5B). The comparison with microspheres in Tris-HCl buffer showed that after 4 h up to 40% of binase was released (Figure 5A), and the complete release of the enzyme occurred after 24 h. Thus, the enzyme released both from the hydrogel pores in Tris-HCl buffer and during the destruction of microspheres in PBS after 4 h was about 50% of the enzyme remains in microcapsules. This result confirms the possibility to prolong the action of encapsulated enzyme.

Since the enzyme therapeutic effect should be manifested in the human gastrointestinal tract, we tested the stability of microcapsules under the model physiological conditions, namely in the phosphate-containing buffer PBS (Figure 5B). Considering the low stability of microspheres in the presence of phosphates, we applied several approaches to prolong the enzyme release time and increase the stability of microcapsules. So, we introduced additional components into alginate microcapsules, namely beta-casein, gelatin, and carbon nanotubes [31,46,47]. It was found that beta-casein binds RNase well (Figure 6A); but when included in microcapsules, the RNase release time increases by 15 min only (Figure 6B). In the case of gelatin included in microcapsules, their stability increased significantly: full integrity was maintained for 24 h. During this time, the release of RNase from microspheres caused the increase in the toxic effect towards tumor cells in comparison with unloaded microspheres (Figure 7); the addition of carbon nanotubes did not change the situation (data not shown).

### 3.4. Cytotoxicity of Alginate Microspheres towards Tumor Cells

The cytotoxicity of drug-free and binase-loaded alginate-based potential delivery systems was studied using duodenal adenocarcinoma HuTu-80 cells during 48 h of incubation. The viability of cells treated by 100 μg/mL binase was reduced by 25%. We showed that the “enzyme-free” alginate hydrogel microspheres were also toxic for HuTu-80 cells, their viability decreased by more than 80%, and the loading of microspheres by binase did not increase the system’s cytotoxicity (Figure 7). The testing of alginate microspheres, stabilized by Ca^2+^, Ni^2+^, Cu^2+^, Zn^2+^, Ba^2+^, and Mn^2+^, towards HuTu-80 cells showed that all studied systems studied significantly reduced cell viability. It can be concluded that the rapid release of divalent ions used for gel preparation from the destroying alginate matrix causes the cytotoxicity of microcapsules. However, an increase in the stability of microcapsules in the presence of gelatin made it possible to fix the toxic effect of RNase itself (Figure 7, Ca/G and Mn/G). Moreover, since the cell growth medium contains less phosphate ions, microcapsules break down more slowly, remaining undestroyed for 48 h. Thus, it can be concluded that optimization of microcapsules composition allows avoiding their destruction and provides a prolonged time of enzyme release.

## 4. Discussion

Microencapsulation is one of the main technologies used in polymer drug delivery systems. Many materials that form nanocarriers with hydrophilic cavities can encapsulate protein cargo. High alginate biocompatibility and the ability to regulate hydrogels stability give grounds for its use as a carrier of therapeutic agents in medicine [48,49]. Alginate particles increasingly demonstrate the effectiveness of drug encapsulation and release, while the maintaining of biological activity of drugs, including proteins, cytokines, and small molecules. Alginate extracted from brown seaweed has been investigated for biomedical and pharmaceutical applications due to its relatively low cost, low toxicity, biocompatibility, and biodegradability. Through the formation of the oil-in-water emulsions and subsequent exposure to divalent cations, alginate particles are created on the micro- or nanometer scale, often referred as ionically cross-linked alginate particles.

The manufacturing of loaded alginate microspheres and sub-microspheres is an interesting approach for drug delivery due to relatively mild gelation process in the presence of ions. The limitations associated with relatively weak ionic bonds result in poor drug encapsulation efficiency and a fast drug release rate (<24 h) [50]. Our results are in accordance with this trend. However, our method of preparing the loaded microspheres allows the enzyme incorporation of about 20 μg into each microsphere. In contrast to the PBS-based system, where the enzyme is completely released within 24 h (Figure 5B), in the cell growth medium the duration of the RNase release from alginate microspheres containing gelatin was at least twice as long (Figure 7). Microcapsules based only on alginate were destroyed much faster, leading to the rapid release of divalent ions in high concentrations and making it impossible to assess the contribution of binase itself to induced cytotoxicity. Simple calculations according to the data in Figure 7 show that if 100 μg/mL of binase causes the death of 30% tumor cells within 48 h, then the decrease in the cell survival induced by loaded microcapsules by 20% (calcium-fortified alginate-gelatin capsules) corresponds to release of 70% RNase. The manganese-fortified alginate–gelatin capsules were more toxic and did not show a significant contribution of released binase to cell death. As can be seen from Figure 7, free ions are toxic to cells. However, the use of gelatin in capsule content increases not only their stability, but also the binding of ions, preventing their release into the cell culture medium and, thereby, maintaining their viability. The interaction of gelatin with divalent ions Co^2+^, Mn^2+^, and Ni^2+^, investigated by fluorescence quenching method, revealed that ions interact with amide bonds in gelatin molecule [51]. We assume that calcium ions interact with gelatin in a similar way.

In cation series, used in the present study, one can detect alterations in the pore size and shape with increasing cation radius [52]. The revealed difference in the internal microstructure of microspheres based on different cations can be of high practical significance due to the possibility for alteration of retention capacity and release of loaded substance. To estimate the enzyme release by means of diffusion from the pores, the Tris-HCl buffer was used, which had no destructive effect on alginate hydrogel. One can see a trend for correlation between the pore dimensions and the RNase release (Figure 5A)—the greater is the pore size, the more effective the binase release. The alternative way for release of drug component (binase) from the alginate matrix can be realized by partial destruction of polysaccharide matrix. In this case, a hydrogel from the ion-crosslinked alginate chains is step-by-step breaking down in the presence of phosphate anion or other ions (sodium, potassium, magnesium), preventing cross-linking of alginate chains [53]. In phosphate buffer, microspheres are subject to destruction. Microscopic analysis showed that the number of intact microspheres with clear borderlines decreased significantly just after 4 h of incubation and amounted to almost zero value after 24 h (Figure 5B).

It is known that metal ion gradients in epidermal tissues serve critical functions in the basal cell proliferation, post-mitotic migration, and functional differentiation in normal homeostasis and in repair following injury. Calcium, zinc, magnesium, and iron are essential trace elements in physiology [54]. Although a decrease in calcium concentration leads to a decrease in the cytotoxic effect on L929 cells in vitro and in full-thickness pig wounds in vivo [55], alginate without calcium ions (or other divalent metals) did not form stable gels. That is, we failed to obtain the unloaded microspheres that are non-toxic to cells. In our experiments, we found that biomedical application of hydrogels can be hindered by toxicity of crosslinking cations. Thus, the use of hydrogels as the carriers of therapeutic drugs in biomedicine is limited because of a number of reasons, which can be very individual in different hydrogel constructions. Therefore, a wide range of diverse studies is extremely important as the preliminary information pool.

Here, we performed a comprehensive study of the properties of the alginate-based drug delivery systems as nanocarriers for a prospective enzymatic drug, RNase. It could be concluded that the combination of alginate with gelatin as a microsphere matrix, fortified with calcium and loaded with RNase, made it possible to reveal the anticancer effect of released enzyme. In conclusion, we note the attractiveness of promising antitumor therapy using binase. The toxic effect of this RNase toward cancer cells is due to expression of specific oncogenes, e.g., KIT, RAS, AML1/ETO, and FLT3.1, whose mutations lead to activation of signaling pathways in various cancer cells and their uncontrolled proliferation [56,57]. In this regard, the possibility of oral administration of the binase in alginate–gelatin microcapsules, providing its prolonged release in gastrointestinal tract, is especially relevant and needs further experimental studies.

## Figures and Tables

**Figure 1 polymers-14-02461-f001:**
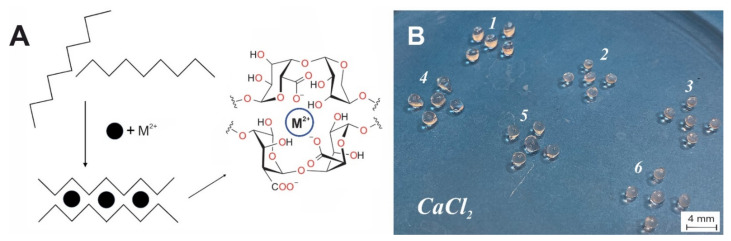
(**A**)—schematic illustration of alginate-based formation according to egg-box model; (**B**)—pure alginate microspheres (1–3) and CNT-modified ones (4–6) stabilized by Ca^2+^.

**Figure 2 polymers-14-02461-f002:**
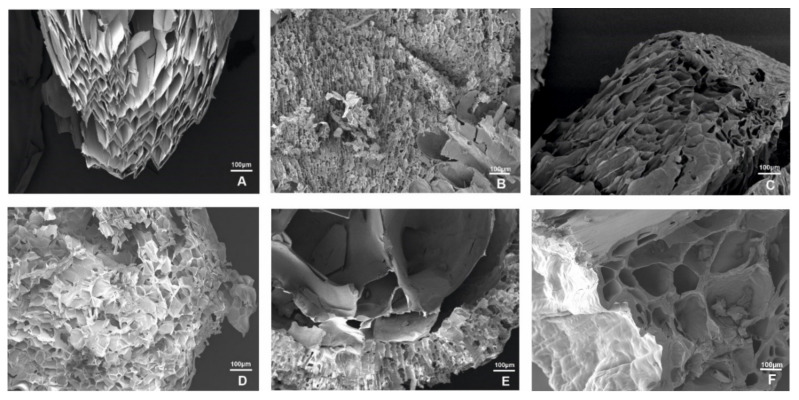
The internal structure of alginate beads stabilized by divalent cations: Ba^2+^ (**A**), Mn^2+^ (**B**), Ca^2+^ (**C**), Zn^2+^ (**D**), Cu^2+^ (**E**), and Ni^2+^ (**F**).

**Figure 3 polymers-14-02461-f003:**
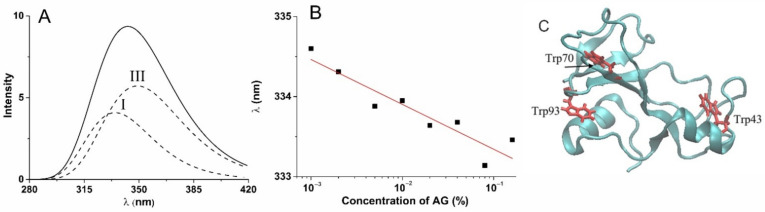
Decomposition of tryptophan fluorescence spectra of binase in alginate hydrogel by two components centered at 334.8 (tryptophan class I) nm and 349.0 nm (tryptophan class III) (**A**); the shift of the tryptophan class I maximum in dependence from alginate concentration (**B**); and binase tertiary structure showing location of three tryptophan residues (**C**).

**Figure 4 polymers-14-02461-f004:**
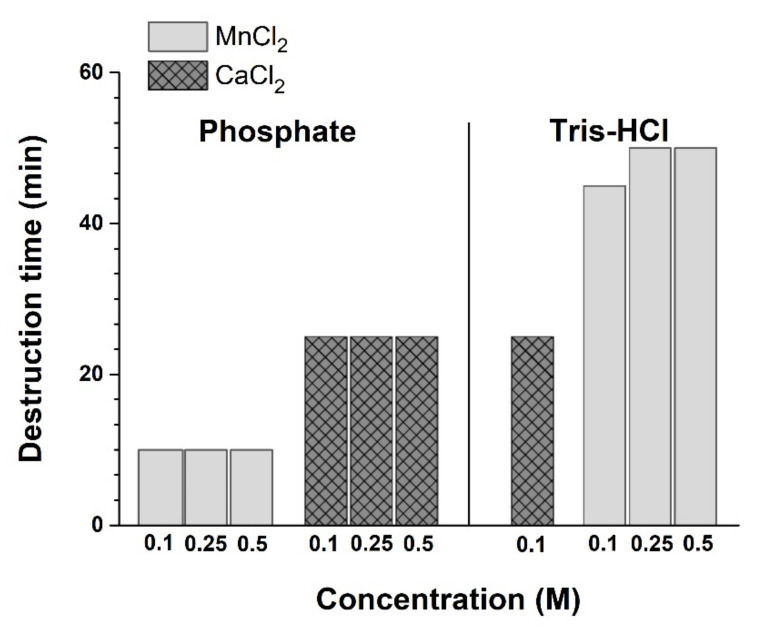
Stability of microcapsules prepared with reduced salt concentration PBS pH 7.2 and Tris-HCl buffer pH 7.4.

**Figure 5 polymers-14-02461-f005:**
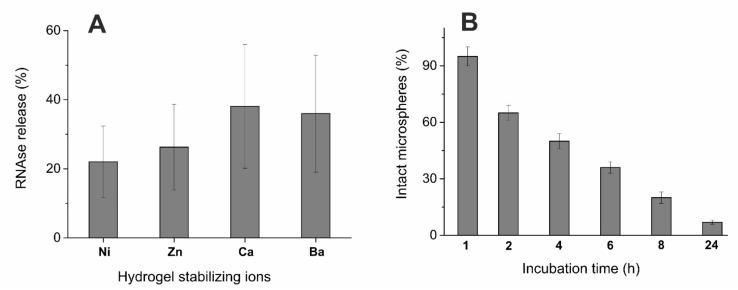
Binase release from cation–alginate microspheres in 50 mM Tris-HCl pH 7.4, 4 h after loading (**A**) and decrease of intact calcium–alginate microspheres proportion in 0.01 M PBS pH 7.2, depending on incubation time (**B**). At panel A, 100% of the amount of enzyme in solution after total microsphere destruction was taken. At panel B, 100% of the number of microspheres visually detected as intact ones, including microspheres with partially injured surfaces, was taken.

**Figure 6 polymers-14-02461-f006:**
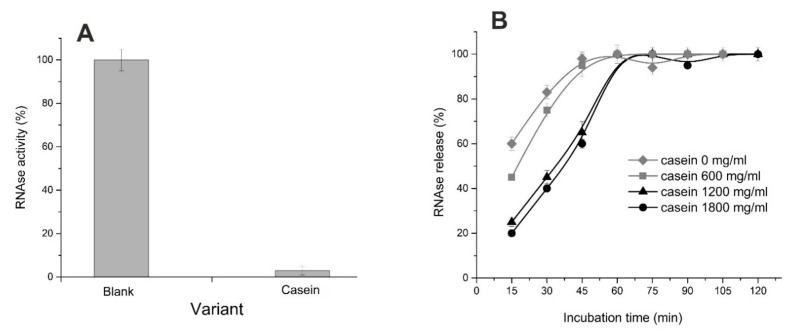
RNase activity in blank solution (300 µg/mL in 0.01 M PBS pH 7.2) and in solution containing 1 mg/mL beta-casein after ultrafiltration through a membrane filter with a pore diameter of 30 kDa (**A**) and time-course of binase release from alginate hydrogel and alginate hydrogel supplemented by beta-casein (**B**). 100% of RNase activity of the loaded protein (calculated as difference between the activity of initial solution in 0.01 M sodium phosphate buffer and resting activity after hydrogels incubation) was taken.

**Figure 7 polymers-14-02461-f007:**
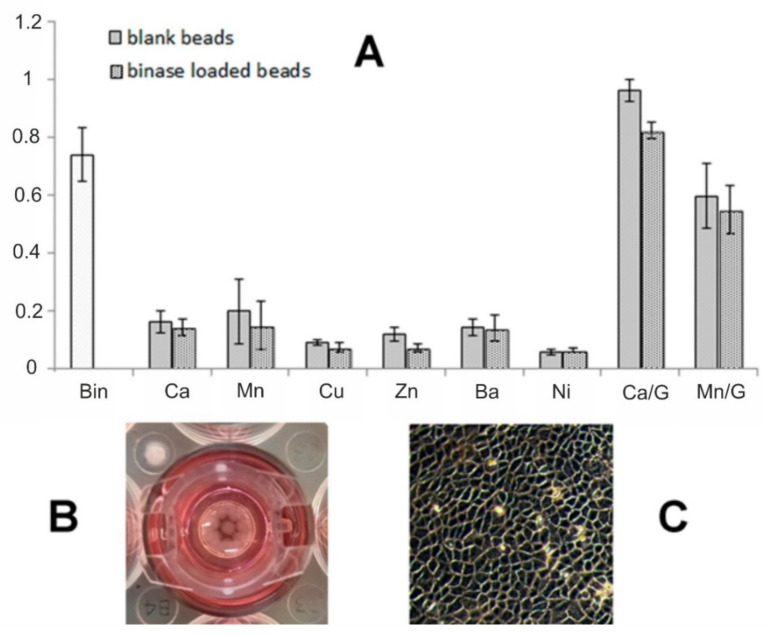
(**A**)—HuTu-80 cells viability after 48 h treatment of alginate microspheres stabilized by divalent cations. Bin—the viability in the presence of 100 μg/mL binase in growth medium. G—ECS with gelatin. (**B**)—top view of the well with cultural insert containing studied microspheres. (**C**)—monolayer of growing HuTu-80 cells in the well (bottom view).

**Table 1 polymers-14-02461-t001:** Water swelling of alginate hydrogels stabilized by divalent cations.

Cation	Water Swelling (%)
Ca^2+^	287
Mn^2+^	408
Ni^2+^	432
Cu^2+^	409
Zn^2+^	250
Ba^2+^	323

## Data Availability

Not applicable.

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
