# Peer review of "Alginate-Based Hydrogel as Delivery System for Therapeutic Bacterial RNase"

_polymers, 2022, doi:10.3390/polym14122461_

Round 1
Reviewer 1 Report
This manuscript reports alginate-based hydrogel as delivery system for therapeutic bacterial RNase. Calcium was used as a biogenic ion strengthening structure of the microspheres. RNase was encapsulated in the hydrogel. The hydrogel had controlled release of RNase. The topic and results are interesting. However, many points should be revised.
1. The scale bar should be added in Fig.1.
2. The authors should give specific names of samples in All Figures and manuscript.
3. The addition of MCNTs should be given the reason.
4. The authors should give more introduction of RNase in the antitumor application. The authors should offer the innovativeness of alginate-based hydrogel as delivery system for therapeutic bacterial RNase.
5. The authors should change the color and other factors to clear illustrate the Fig. 4.
6. The results in Fig.7 are confusing. The testing of alginate microspheres, stabilized by Ca2+, Ni2+, Cu2+,Zn2+, Ba2+, Mn2+, towards HuTu-80 cells shows that all studied systems studied significantly reduce cell viability, but the positive group had no significant effect on tumor cells compared with the control group. More importantly,the inhibitory effect of Ca/G and Mn/G on tumor cells is significantly reduced, please give a reasonable explanation.
7. The method of determination the concentration of RNase should be added. The method of calculation of loading content of RNase should be added.
Author Response
Dear Reviewer!
Thank you for the interest to our manuscript. We thank you for constructive and helpful comments and revised the manuscript accordingly to your questions and suggestions. Below we address each of the points raised.
We prepared the corrected version of our manuscript. Our answers are after words “Author’s reply”. Below we added new text fragments in the manuscript colored in red. Also, there are red inserts in the revised manuscript.
- The scale bar should be added in Fig.1.
Author’s reply: Done.
- The authors should give specific names of samples in All Figures and manuscript.
Author’s reply: The only samples for which we could give specific names were metal ions. We have added chemical symbols when it was necessary. What about our enzyme, we designated it in section Ribonuclease in Materials and Methods and used its traditional name binase.
- The addition of MCNTs should be given the reason.
Author’s reply: We include additional text fragment in the end of Introduction:
The employment of polysaccharides in real biomedical practice often requires ad-ditional modifications of engineered systems. One of the accepted ways to improve functional properties of desired systems is the use of such nanomaterials as carbon nanotubes (CNTs) [31].
- The authors should give more introduction of RNase in the antitumor application. The authors should offer the innovativeness of alginate-based hydrogel as delivery system for therapeutic bacterial RNase.
Author’s reply: We include additional text into Introduction:
Many of RNases attack selectively malignant cells, triggering apoptotic response, and therefore they are considering as alternative chemotherapeutic drugs. Unlike mammalian RNases, being potently inhibited by ubiquitous cytoplasmic eukaryotic RNase inhibitor, the guanyl-preferring secreted RNase from Bacillus pumilus (binase) manifests its catalytic activity towards available RNA molecules inside cells as a result of their high stability and lack of susceptibility to RNase inhibitor. For loading into microspheres, we have chosen binase possessing antiviral [25-27] and antitumor [28-30] properties. The killer strategy of cytotoxic RNases includes the hitting of its main target, RNA, but does not exclude additional ravages leading to the cell death [28]. It was shown that binase is able to inhibit the replication of MERS-CoV and of the low-pathogenic human coronavirus 229E in cell culture, decreasing titers of both viruses at non-toxic concentrations, reducing accumulation of viral subgenomic RNAs, viral nucleocapsid protein and non-structural protein 13 [25]. Moreover, binase acts as an antiviral agent at the level of whole animal organism infected by mammalian orthoreovirus 1 strain Lang [27]. The selective tumor cytotoxicity of binase is determined not only by its catalytic activity and the ability to avoid inhibition by mammalian RNase inhibitor, but also by its ability to interact with certain cellular components and oncogenic proteins [29]. Considering our obtained results, it is necessary to add moreover a change in the profile of cellular microRNAs. The analysis of binase-susceptible miRNAs and their regulatory networks showed that the main modulated events were the transcription and translation control, the cell cycle, cell proliferation, adhesion and invasion, apoptosis and autophagy, as well as some other tumour-related cascades, with an impact on the observed anti-tumour effects [30]. These data give a reasonable idea of the binase use as the antitumor agent. Despite the enzyme resistance to proteolysis, its oral administration in encapsulated form will create favorable conditions for gradual release in the gastrointestinal tract. Alginate is known as biocompatible harmless substance [8,13] and is used in medicine to bind and remove from the body the metabolic by-products, salts of heavy metals and radionuclides as well as to reduce stomach acidity (for ex., Antacid from USA manufacturers, NutraScience Labs, Pharbest Pharmaceuticals, NY). Since alginate neutralizes the acidic environment, it will contribute to a better manifestation of binase catalytic activity, whose pH-optimum is 8.5. That is why we set as the goal of this work to include binase in the alginate-based hydrogel.
- The authors should change the color and other factors to clear illustrate the Fig. 4.
Author’s reply: Done.
- The results in Fig.7 are confusing. The testing of alginate microspheres, stabilized by Ca2+, Ni2+, Cu2+,Zn2+, Ba2+, Mn2+, towards HuTu-80 cells shows that all studied systems studied significantly reduce cell viability, but the positive group had no significant effect on tumor cells compared with the control group. More importantly, the inhibitory effect of Ca/G and Mn/G on tumor cells is significantly reduced, please give a reasonable explanation.
Author’s reply: We modified Fig. 7 and added additional explanation to Discussion.
As can be seen from Figure 7, free ions are toxic to cells. However, the use of gelatin in capsule content increases not only their stability, but also the binding of ions, preventing their release into the cell culture medium and, thereby, maintaining their viability. The interaction of gelatin with divalent ions Co2+, Mn2+ and Ni2+, investigated by fluorescence quenching method, revealed that ions interact with amid bond in gelatin molecule [51]. We assume that calcium ions interact with gelatin in a similar way.
- The method of determination the concentration of RNase should be added. The method of calculation of loading content of RNase should be added.
Author’s reply: Subsection of Materials and Methods “Enzyme release from hydrogel” has been extended to “Enzyme release from hydrogel and RNase content in microspheres”
Enzyme release from beta-casein or gelatin-containing alginate microspheres was measured by the RNase catalytic activity.
To determine activity of RNase, loaded into microspheres without gelatin, the protein concentration was measured spectrophotometrically at 280 nm after complete lysis of 10 capsules in 10 ml PBS. Protein concentration in 1 ml solution corresponds to content of RNase in one microsphere. For gelatin-containing microspheres, catalytic activity of RNase was determined in this solution, and the activity units were recalculated for content of protein in one microsphere.

Reviewer 2 Report
Dear author, please revise your manuscript to the following suggested points
I strongly recommend revising this manuscript as follows:
1. I would like to recommend the authors, to enhance the reader's understanding please include a schematic illustration with real hydrogel and chemical structure.
2. In the introduction part of this manuscript author should cite the following latest articles: A. Postoperative Adhesion Barriers." Macromolecular bioscience 21, no. 3 (2021): 2000395. B. Balance of antiperitoneal adhesion, hemostasis, and operability of compressed bilayer ultrapure alginate sponges. Biomaterials Advances, 212825.
3. Author should compare the mechanical strength of various ions in the crosslinked hydrogels through a Rheometer or other appropriate techniques. authors can follow the following articles A.Dually crosslinked injectable hydrogels of poly (ethylene glycol) and poly [(2-dimethylamino) ethyl methacrylate]-b-poly (N-isopropyl acrylamide) as a wound healing promoter." Journal of Materials Chemistry B 5, no. 25 (2017): 4955-4965.B. Self-assembly of partially alkylated dextran-graft-poly [(2-dimethylamino) ethyl methacrylate] copolymer facilitating hydrophobic/hydrophilic drug delivery and improving conetwork hydrogel properties." Biomacromolecules 19, no. 4 (2018): 1142-1153.
4. Author should compare the in-vitro degradation rate of various ions the crosslinked hydrogels, author can follow above articles.
5. Author should compare the equilibrium swelling capacity of various ions the crosslinked hydrogels, the author can follow articles. Effect of polyethene glycol on properties and drug encapsulation–release performance of biodegradable/cytocompatible agarose–polyethene glycol–polycaprolactone amphiphilic co-network gels." ACS applied materials & interfaces 8, no. 5 (2016): 3182-3192, A. Reactive compatibilizer mediated precise synthesis and application of stimuli-responsive polysaccharides-polycaprolactone amphiphilic co-network gels." Polymer99 (2016): 470-479.
6. Author should add the cell images of the in-vitro cytocompatibility assay in the revised manuscript.
7. Author should check the grammar and typo errors carefully in the revised manuscript
Author Response
Dear Reviewer!
Thank you for the interest to our manuscript. We thank you for constructive and helpful comments and revised the manuscript accordingly to your questions and suggestions. Below we address each of the points raised.
We prepared the corrected version of our manuscript. Our answers are after words “Author’s reply”. Below we added new text fragments in the manuscript colored in red. Also, there are red inserts in the revised manuscript.
- I would like to recommend the authors, to enhance the reader's understanding please include a schematic illustration with real hydrogel and chemical structure.
Author’s reply: We included schematic illustration of hydrogel formation and chemical structure in Figure 1.
- In the introduction part of this manuscript author should cite the following latest articles: A. Postoperative Adhesion Barriers." Macromolecular bioscience 21, no. 3 (2021): 2000395. B. Balance of antiperitoneal adhesion, hemostasis, and operability of compressed bilayer ultrapure alginate sponges. Biomaterials Advances, 212825.
Author’s reply: We have included the recommended references.
- Author should compare the mechanical strength of various ions in the crosslinked hydrogels through a Rheometer or other appropriate techniques. authors can follow the following articles A.Dually crosslinked injectable hydrogels of poly (ethylene glycol) and poly [(2-dimethylamino) ethyl methacrylate]-b-poly (N-isopropyl acrylamide) as a wound healing promoter." Journal of Materials Chemistry B 5, no. 25 (2017): 4955-4965. B. Self-assembly of partially alkylated dextran-graft-poly [(2-dimethylamino) ethyl methacrylate] copolymer facilitating hydrophobic/hydrophilic drug delivery and improving conetwork hydrogel properties." Biomacromolecules 19, no. 4 (2018): 1142-1153.
Author’s reply: Dear Reviewer, thank you for interesting suggestions. Unfortunately, our present paper is devoted to biomedical aspects of studied systems. Indeed, the mechanical strengthening of hydrogels is very interesting for us too, but not in this rather compact work.
- Author should compare the in-vitro degradation rate of various ions the crosslinked hydrogels, author can follow above articles.
Author’s reply: Thank you. We shell make the suggested experiments in our future work.
- Author should compare the equilibrium swelling capacity of various ions the crosslinked hydrogels, the author can follow articles. Effect of polyethene glycol on properties and drug encapsulation–release performance of biodegradable/cytocompatible agarose–polyethene glycol–polycaprolactone amphiphilic co-network gels." ACS applied materials & interfaces 8, no. 5 (2016): 3182-3192, A. Reactive compatibilizer mediated precise synthesis and application of stimuli-responsive polysaccharides-polycaprolactone amphiphilic co-network gels." Polymer99 (2016): 470-479.
Author’s reply: we have added additional information to Section Materials and Methods (Determination of swelling)
Water swelling capacity of microspheres
The swelling rate was determined by the known method [37-39]. A weighted quantity of lyophilized microspheres was submerged into 50mM Tris-HCl buffer (pH 7.4) at a fixed temperature (25°C) in a shaker (Mikrowstrzasarka ML-1) at shaking speed 100 rpm. Microspheres were removed from solution after 24 h; excess water was wiped off and the weight of swollen microspheres was determined. The swelling rate was determined using the following expression:
Swelling (%)=(m_s-m_d)/m_d ×100, (1)
where ms and md are the masses of water-swollen and dry microspheres, respectively.
and add information to Section 3.1.1.
The water swelling of microspheres is a result of solvent absorption by hydrogel structure. This parameter correlates with stability of hydrogel. Table I shows the value of water swelling of microspheres stabilized with various cations. According to our data the lowest water swelling was observed for microspheres stabilized with Ca2+ and Zn2+.
- Author should add the cell images of the in-vitro cytocompatibility assay in the revised manuscript.
To Figure 7 images of well inserts and growing cells were added (7B and 7C, respectively).
Author’s reply: Dear Reviewer, we modified Figure 7 according to your comments.
- Author should check the grammar and typo errors carefully in the revised manuscript.
Author’s reply: Thank you, we made it.

Round 2
Reviewer 1 Report
The authors have addressed the problem very well, and the manuscript can be accepted in the present form.
Reviewer 2 Report
AUthors has revised all suggested points